# Antagonistic Interactions between Dicamba and Glyphosate on Barnyardgrass (*Echinochloa crus-galli*) and Horseweed (*Erigeron canadensis*) Control

Estefania G. Polli *, Leandro H. S. Guimaraes, Jose H. S. de Sanctis and Greg Kruger 

Department of Agronomy and Horticulture, University of Nebraska-Lincoln, North Platte, NE 69101, USA
* Correspondence: egomier@ncsu.edu; Tel.: +1-(919)-909-3618

**Abstract:** Dicamba plus glyphosate (DpG) tank mixture has been widely adopted for broad-spectrum weed control. However, recent studies indicated antagonistic interactions between these herbicides. Furthermore, little is known about the effect of non-ammonium sulfate water conditioner (non-AMS WC) adjuvant on the activity of DpG tank mixture. Thus, the present study was elaborated to evaluate (1) the interaction of DpG tank mixtures combinations on barnyardgrass, and glyphosate-susceptible (GS) and -resistant (GR) horseweed control, and (2) the effect of non-ammonium sulfate water conditioner (non-AMS WC) on the interaction of those two herbicides on the aforementioned weed species. Greenhouse experiments were conducted in 2020 at the Pesticide Application Laboratory in North Platte, NE. Herbicide treatments were arranged in a two-level factorial design of six dicamba rates by six glyphosate rates for Study 1, and in a three-level factorial design of two adjuvant treatments (presence or not of non-AMS WC) by four dicamba rates by four glyphosate rates for Study 2. Both trials were conducted as randomized complete block designs with four replications and two runs. Antagonistic interactions were observed throughout DpG treatments in GS and GR horseweed. For instance, dicamba (560 g ae ha$^{-1}$) and glyphosate (1260 g ae ha$^{-1}$) resulted in 72% of GR horseweed biomass reduction, compared to 81% of estimated biomass reduction. For barnyardgrass, antagonistic interactions were only observed within the reduced glyphosate rates. The addition of non-AMS WC had no effect on DpG antagonism. However, it improved the barnyardgrass control by glyphosate.

**Keywords:** glyphosate-resistant; horseweed; barnyardgrass; glyphosate; dicamba; non-ammonium sulfate water conditioner; antagonism

## 1. Introduction

Glyphosate, the most commonly used herbicide in the United States, was first registered as a herbicide in 1974. For the first two decades after glyphosate introduction to the market, its use was restricted to broad-spectrum weed control in burndown and noncrop applications [1]. In 1996, glyphosate-tolerant technology was introduced to the market, first available in soybeans (*Glycine max* (L.) *Merr*.), followed by corn (*Zea mays* L.), cotton (*Gossypium hirsutum* L.), canola (*Brassica napus* L.), and sugar beets (*Beta vulgaris* L.). This technology widely expanded glyphosate use to postemergence applications throughout the growing season. However, the continuous use of this herbicide associated with the lack of herbicide mode of action (MOA) rotation resulted in the evolution of glyphosate resistance by several weed species. Currently, there are 17 glyphosate-resistant (GR) weed species reported in the United States [2].

In response to the evolution of GR weeds, dicamba-tolerant (DT) cotton and soybean crops, which are also tolerant to glyphosate, were launched in the market in 2016 and 2017, respectively, to provide an alternative MOA to glyphosate-tolerant cropping systems [3]. The DT technology was quickly adopted by growers: approximately 43% and 69% of soybean and cotton acreages in the U.S were planted with DT seeds in 2018 and 2019,

respectively [4,5]. Hence, tank mixing dicamba and glyphosate has become a common practice for broad-spectrum and herbicide-resistant weed management in soybean and cotton fields.

The addition of glyphosate to dicamba tank mixtures can enhance the control of various broadleaf species and suppress herbicide-resistant biotypes [6–8]. However, previous studies reported that combinations of these two herbicides can have an antagonistic effect on the control of velvetleaf (*Abutilon theophrasti Medik.*) [9], giant ragweed (*Ambrosia trifida* L.) [10], kochia (*Bassia scoparia* (L.) A. J. Scott) [11], and johnsongrass (*Sorghum halepense* (L.) Pers.) [12], which suggests that the nature of this interaction is species-specific.

In a 2019 Weed Science Society of America (WSSA) survey, horseweed (*Erigeron canadensis* L.) ranked as the second and third most troublesome weed species in soybean and cotton fields, respectively [13]. In the same survey, barnyardgrass (*Echinochloa crus-galli* (L.) P. Beauv.) ranked as the fourth most troublesome weed in cotton fields. Despite the importance of these two weed species in DT crops, little is known about the interaction of dicamba and glyphosate on horseweed and barnyardgrass control. Additionally, limited information is available about the effect of adjuvants on DpG interaction nature.

According to Penner [14], adjuvants can impact herbicide antagonism in at least two ways: (1) directly increasing herbicide absorption and (2) preventing the formation of herbicidal forms with reduced absorption. Dicamba and glyphosate are weak acid herbicides that can react to cationic salts in the water, and form molecular complexes that are less likely to be absorbed and moved within the plant [15]. Previous studies demonstrated that the addition of ammonium sulfate (AMS) to DpG solutions is beneficial in overcoming salt antagonism and increasing herbicidal efficacy in several weed species [16–19]. However, the use of AMS is restricted for dicamba application since this adjuvant increases dicamba vapor formation [20–22]. Non-AMS water conditioners are an alternative to minimizing the negative effect of salt antagonism in hard water for DpG tank mixtures without increasing potential dicamba volatilization. Zollinger et al. [23] reported that the control of 17 broadleaf and grass species by DpG solutions containing non-AMS WC was similar to that of solutions containing AMS when applied to 1000 ppm hard water. Therefore, the objectives of this study were to evaluate the nature of interaction between dicamba and glyphosate on barnyardgrass, and glyphosate-susceptible (GS) and -resistant horseweed control, and the effect of non-ammonium sulfate water conditioner on this interaction on barnyardgrass and GR horseweed control.

## 2. Material and Methods

### 2.1. Location, Plant Material, and Application Information

Greenhouse studies were conducted at the Pesticide Application Technology Laboratory in North Platte, NE from July to December 2020.

Barnyardgrass, GS horseweed, and GR horseweed seeds were sown on 20 July and 12 August for the first and second runs of Study 1, respectively, and on 9 and 22 November for the first and second runs of Study 2, respectively. Barnyardgrass seeds were sown and grown in RLC4 containers (Stuewe and Sons Inc., Corvallis, OR, USA) containing a potting mix (Pro-mix BX5, Premier Tech Horticulture Ltd., Riviere-du-Loup, Canada). Horseweed seeds were first sowed in a 29.8 × 23.8 × 5 cm aluminum pan (Mainstays™, Walmart®, Bentonville, AR, USA) containing the previously mentioned potting mix. A single pan was used for each horseweed population. Once horseweed plants had reached the 3-leaf stage, each plant was transplanted to a single RLC4 container.

Plants were maintained under controlled conditions of 30/20 °C (day/night), 60% ± 10% RH, and 16 h of daylight (Philips Lighting, Somerset, NJ, USA). Additionally, plants were watered daily and fertilized (UNL 5-1-4, Wilbur-Ellis Agribusiness, Aurora, CO, USA), and treated for loopers and other insects (Gnatrol WDG®, Valent U.S.A., Walnut Creek, CA, USA) as needed. The glyphosate-susceptible and -resistant horseweed populations were collected in Jefferson and Wheeler counties in Nebraska, respectively, and previously submitted to dose–response bioassays to confirm susceptibility or resistance to glyphosate [24].

Applications were performed when barnyardgrass plants were 15 cm tall and horseweed rosettes were 10 cm wide using a three-nozzle spray chamber (Generation III Research Track Sprayer DeVries Manufacturing, Hollandale, MN, USA), calibrated to deliver 140 L ha$^{-1}$ through TT11002 nozzles (TeeJet Technologies Spraying Systems Co., Glendale Heights, IL, USA) at operating pressure of 276 kPa and constant speed of 6.5 km h$^{-1}$.

### 2.2. Dicamba and Glyphosate Tank Mixture Interactions (Study 1)

The study was organized as a complete block design containing 4 replications and 2 experimental runs. Each individual plant was considered to be one replication. Treatments were arranged in a two-level factorial design of 6 dicamba (Xtendimax ® with Vapor Grip®, Monsanto Company, St. Louis, MO, USA) rates (0, 140, 280, 560, 840, and 1118 g ae ha$^{-1}$) by 6 glyphosate (Roundup PowerMax®, Monsanto Company, St. Louis, MO, USA) rates (0, 316, 630, 1260, 1890, and 2520 g ae ha$^{-1}$). Herbicide rates corresponded to 0x, 0.25x, 0.5x, 1x, 1.5x, and 2x of the recommended field dose. Plants were cut above the ground 28 days after application (DAA) and oven-dried at 65 °C until constant dry weight. Dry weight data were recorded and converted into percentages of biomass reduction as compared with the nontreated control according to Equation (1) [25]:

$$BR = 100 - \frac{(X * 100)}{Y} \tag{1}$$

where BR is the biomass reduction (%), X is the dry weight (g) of an individual experimental unit, and Y is the mean biomass (g) of the nontreated control replicates.

### 2.3. Effect of Non-Ammonium Sulfate Water Conditioner on Dicamba and Glyphosate Tank Mixture Interactions (Study 2)

Similar to Study 1, Study 2 was organized as a complete block design containing 4 replications and 2 experimental runs. An individual plant was considered one replication as well. Treatments were arranged in a three-level factorial design of 4 dicamba (Xtendimax® with Vapor Grip®, Monsanto Company, St. Louis, MO, USA) rates (0, 140, 280, 560 g ae ha$^{-1}$) by 4 glyphosate (Roundup PowerMax®, Monsanto Company, St. Louis, MO, USA) rates (0, 316, 630, 1260 g ae ha$^{-1}$), and the presence or not of non-ammonium sulfate water conditioner (Class Act® Ridion®, Winfield Solutions LLC, St. Paul, MN, USA) at 1 v v$^{-1}$. Analyses indicated the presence of 188 mg L$^{-1}$ of CaCO$_3$ in the tap water collected from the laboratory. According to USGS (2020) [26], water containing more than 180 mg L$^{-1}$ is categorized as very hard. Additionally, DRA (IntactTM, Precision Laboratories LLC, Waukegan, IL, USA) at 0.5 v v$^{-1}$ and VRA (VapexTM, Kalo Inc, Overland Park, KS, USA) at 1286 mL ha$^{-1}$ were added to all treatment solutions due to new regulations involving dicamba applications. Only the four lowest rates (0x, 0.25x, 0.5x, and 1x) of each herbicide from Study 1 were selected for this study because of the high sensibility of plants to 1.5x and 2x rates. Furthermore, due to the high mortality of GS horseweed in Study 1, only barnyardgrass and GR horseweed were used in Study 2. Data collection was performed exactly as described in Study 1.

### 2.4. Statistical Analyses

Data were subjected to ANOVA to test the significance of fixed and random effects. Replications and runs were treated as random effects, and non-AMS WC, dicamba rates, and glyphosate rates as fixed effects. Mean separation procedures were conducted with Tukey's least-significant-difference test at $\alpha$ = 0.05. Statistical analyses were performed in R software utilizing the base packages.

Gowing's correlation analysis was used to evaluate the type of dicamba and glyphosate interactions. Expected biomass reduction values for the different herbicide interactions were calculated using Equation (2) [27,28]

$$E = (X + Y) - (XY/100) \tag{2}$$

where *E* is the expected biomass reduction (%) with dicamba and glyphosate in the tank mixture at a specific rate, and X and Y are the observed biomass reduction (%) with the application of dicamba and glyphosate alone, respectively, at the same specific rates. The expected and observed biomass reduction values for DpG tank mixtures were subjected to *t*-tests to determine whether means differed at $\alpha = 0.05$. The herbicide combination was considered to be antagonistic if the expected mean were significantly greater than the observed mean. If the expected mean were significantly lower than the observed mean, the herbicide combination was considered to be synergistic. When means were not significantly different, the nature of herbicide mixture was considered to be additive [9,27].

## 3. Results

### 3.1. Dicamba and Glyphosate Tank Mixture Interactions (Study 1)

A significant dicamba by glyphosate interaction was observed for barnyardgrass ($p = 0.0253$ and $f = 1.6977$), and GS ($p < 0.0001$ and $f = 14.1173$) and GR horseweed ($p = 0.0002$ and $f = 2.4917$) biomass reduction. Therefore, a two-way ANOVA table is presented for each population and/or species.

### 3.1.1. Barnyardgrass

Due to the reduced efficacy of dicamba on grass species, dicamba applied alone resulted in <28% biomass reduction for all rates. In contrast, glyphosate was highly effective on barnyardgrass, and all rates above 630 g ae ha$^{-1}$ resulted in >88% biomass reduction (Table 1). Further, when glyphosate was applied alone at 316 g ae ha$^{-1}$, barnyardgrass biomass reduction was 81%, and the addition of dicamba at any rate resulted in an antagonistic response that led to a reduction in biomass ranging from 12% to 32% when compared to the expected value. For example, glyphosate at 316 g ae ha$^{-1}$ and dicamba at 560 g ae ha$^{-1}$ resulted in 57% biomass reduction compared to 86% from expected value.

**Table 1.** Effect of dicamba and glyphosate tank mixtures on barnyardgrass biomass reduction (observed and expected) at 28 days after treatment (DAT). Herbicides were applied to 15 cm tall plants.

| Dicamba Dose (g ae ha$^{-1}$) | Glyphosate Dose (g ae ha$^{-1}$) [a,b,c] | | | | | | | | | | | |
|---|---|---|---|---|---|---|---|---|---|---|---|---|
| | 0 | | 316 | | 630 | | 1260 | | 1900 | | 2530 | |
| | Obs | Exp | Obs | Exp | Obs | Exp | Obs | Exp | Obs | Exp | Obs | Exp |
| | | | | | % of Biomass Reduction | | | | | | | |
| 0 | - | - | 81 Ab | - | 95 a | - | 97 a | - | 97 a | - | 97 a | - |
| 140 | 21 c | - | 73 ABb | 85 * | 88 a | 96 | 97 a | 97 | 97 a | 98 | 98 a | 98 |
| 280 | 19 c | - | 57 BCb | 85 * | 97 a | 96 | 98 a | 98 | 98 a | 98 | 97 a | 97 |
| 560 | 23 c | - | 57 BCb | 86 * | 94 a | 96 | 97 a | 98 | 98 a | 98 | 98 a | 98 |
| 840 | 17 c | - | 52 Cb | 84 * | 94 a | 96 | 98 a | 97 | 99 a | 98 | 98 a | 97 |
| 1120 | 28 c | - | 59 BCb | 86 * | 89 a | 96 * | 96 a | 98 | 97 a | 98 | 99 a | 98 |

[a] Different letters indicate significant differences ($p < 0.05$) according to Tukey's least-significant-difference test. Uppercase letters represent differences between dicamba rates at the same glyphosate dose (vertical), and lowercase letters indicate differences between glyphosate rates at the same dicamba dose (horizontal). [b] Expected value obtained from Gowing's equation: E = (X + Y) − (XY/100), where E is the expected percentage control with dicamba and glyphosate in the tank mixtures, and X and Y are the observed percent control with herbicide dicamba or glyphosate alone, respectively. [c] Abbreviations: Obs, observed biomass reduction; Exp, expected biomass reduction according to Gowing's equation. * Significantly different from the observed value ($p < 0.05$) as determined with the *t* test, indicating antagonistic interactions of herbicides applied in tank mixtures.

### 3.1.2. Glyphosate-Susceptible Horseweed

Dicamba was effective in GS horseweed biomass reduction, which varied from 73% to 86% for all rates of dicamba applied alone (Table 2). Glyphosate-alone treatments had a higher variation, and biomass reduction ranged from 17% to 89%. Biomass reduction of DpG tank mixtures varied from 77% to 92%. However, antagonistic interactions were observed throughout most of the tested tank mixture combinations, with decreases in biomass reduction varying from 4% to 21% when compared to expected values obtained

from Gowing's equation. For example, when dicamba and glyphosate were applied at 560 and 1260 g ae ha$^{-1}$, respectively, observed biomass reduction was 82%, compared to 93% from the expected biomass reduction.

Further, two-way ANOVA demonstrated a positive response to increasing dicamba rates at a fixed glyphosate rate. For example, when glyphosate was applied at 630 g ae ha$^{-1}$, the addition of 140 g ae ha$^{-1}$ of dicamba resulted in 59% biomass reduction, compared to 77% when dicamba was added at 1120 g ae ha$^{-1}$. Nonetheless, this behavior was not consistently observed for the increments of glyphosate at a fixed dicamba rate. Instead, when dicamba was applied at 140, 280, and 1120 g ae ha$^{-1}$, no significant effect of glyphosate increments was observed. Furthermore, dicamba applications at 560 and 840 g ae ha$^{-1}$ without glyphosate resulted in 86% and 81% biomass reduction, compared to 73% and 77% when glyphosate was added at 316 g ae ha$^{-1}$ respectively. Obtained results suggest that the interaction of dicamba and glyphosate is antagonistic in GS horseweed where the presence of glyphosate reduces glyphosate activity.

**Table 2.** Effect of dicamba and glyphosate tank mixtures on glyphosate-susceptible horseweed biomass reduction (observed and expected) at 28 days after treatment (DAT). Herbicides were applied to 10 cm diameter plants.

| Dicamba Dose (g ae ha$^{-1}$) | Glyphosate Dose (g ae ha$^{-1}$) [a,b,c] | | | | | | | | | | | |
|---|---|---|---|---|---|---|---|---|---|---|---|---|
| | 0 | | 316 | | 630 | | 1260 | | 1900 | | 2530 | |
| | Obs | Exp | Obs | Exp | Obs | Exp | Obs | Exp | Obs | Exp | Obs | Exp |
| | % of Biomass Reduction | | | | | | | | | | | |
| 0 | - | - | 17 Bd | - | 28 Cd | - | 54 Bb | - | 86 Aa | - | 89 Aba | - |
| 140 | 73 B | - | 77 A | 77 | 70 B | 81 * | 76 A | 88 * | 75 B | 96 * | 82 B | 97 * |
| 280 | 83 A | - | 81 A | 85 * | 73 AB | 88 * | 81 A | 92 * | 84 A | 97 * | 83 B | 98 * |
| 560 | 86 Aa | - | 73 Ac | 88 * | 76 Abc | 90 * | 82 Aab | 93 * | 85 Aa | 98 * | 86 Aa | 98 * |
| 840 | 81 Abc | - | 77 Ac | 84 * | 81 Abc | 86 * | 83 Abc | 91 * | 85 Ab | 97 * | 92 Aa | 98 |
| 1120 | 81 A | - | 80 A | 84 | 83 A | 86 | 84 A | 91 * | 84 A | 97 * | 89 AB | 98 * |

[a] Different letters indicate significant differences ($p < 0.05$) according to Tukey's least-significant-difference test. Uppercase letters represent differences between dicamba rates at the same glyphosate dose (vertical), and lowercase letters indicate differences between glyphosate rates at the same dicamba dose (horizontal). [b] Expected value obtained from the Gowing's equation: E = (X + Y) − (XY/100), where E is the expected percentage control with dicamba and glyphosate in tank mixtures, X and Y are the observed percent control with dicamba or glyphosate alone, respectively. [c] Abbreviations: Obs, observed biomass reduction; Exp, expected biomass reduction according to Gowing's equation. * Significantly different from the observed value ($p < 0.05$) as determined with the *t* test, indicating antagonistic interactions of herbicides applied in tank mixtures.

### 3.1.3. Glyphosate-Resistant Horseweed

GR horseweed biomass reduction when dicamba or glyphosate was applied alone ranged from 50% to 71% and 27% to 54%, respectively (Table 3), and when the tank was mixed, biomass reduction ranged from 59% to 77%.

Antagonistic interactions between dicamba and glyphosate were more consistently observed at higher glyphosate rates, and decreases in biomass reductions ranged from 7% to 21% when compared to expected values. For instance, when glyphosate was applied at 1260 g ae ha$^{-1}$, antagonism was observed for all dicamba rates, with up to 21% decrease in biomass reduction when compared to the expected value.

Similar to GS horseweed, the addition of dicamba increased GR horseweed biomass reduction at a fixed glyphosate rate. However, the increments of glyphosate at the fixed dicamba rates of 560, 840, and 1120 g ae ha$^{-1}$ did not increase efficacy. Moreover, when dicamba was applied alone at 280 g ae ha$^{-1}$, biomass reduction was 65% and the addition of glyphosate at 1260 g ae ha$^{-1}$ decreased biomass reduction to 59%.

**Table 3.** Effect of dicamba and glyphosate tank mixtures on glyphosate-resistant horseweed biomass reduction (observed and expected) at 28 days after treatment (DAT). Herbicides were applied to 10 cm diameter plants.

| Dicamba Dose (g ae ha$^{-1}$) | Glyphosate Dose (g ae ha$^{-1}$) [a,b,c] | | | | | | | | | | | |
|---|---|---|---|---|---|---|---|---|---|---|---|---|
| | 0 | | 316 | | 630 | | 1260 | | 1900 | | 2530 | |
| | Obs | Exp | Obs | Exp | Obs | Exp | Obs | Exp | Obs | Exp | Obs | Exp |
| | % of Biomass Reduction | | | | | | | | | | | |
| 0 | - | - | 27 Dc | - | 37 Dbc | - | 44 Cab | - | 54 Ba | - | 51 Ba | - |
| 140 | 50 Bc | - | 66 Cab | 63 | 59 Cb | 68 | 59 Bb | 72 * | 67 Aab | 77 * | 75 Aa | 75 |
| 280 | 65 Aab | - | 67 Ba | 74 * | 64 Bab | 78 * | 59 Bb | 80 * | 70 Aa | 84 * | 70 Aa | 83 * |
| 560 | 67 A | - | 75 AB | 76 | 73 AB | 79 | 72 AB | 81 * | 73 A | 85 * | 75 A | 84 |
| 840 | 71 A | - | 76 A | 79 | 72 AB | 82 * | 74 A | 84 * | 72 A | 87 * | 77 A | 86 * |
| 1120 | 71 A | - | 74 ABC | 79 | 77 A | 81 | 76 A | 84 * | 72 A | 87 * | 73 A | 86 * |

[a] Different letters indicate significant differences ($p < 0.05$) according to Tukey's least-significant-difference test. Uppercase letters represent differences between dicamba rates at the same glyphosate dose (vertical) and lowercase letters indicate differences between glyphosate rates at the same dicamba dose (horizontal). [b] Expected value obtained from the Gowing's equation: $E = (X + Y) - (XY/100)$, where E is the expected percent control with dicamba and glyphosate in tank mixture, X and Y are the observed percentage control with dicamba or glyphosate alone, respectively. [c] Abbreviations: Obs, observed biomass reduction; Exp, expected biomass reduction according to Gowing's equation. * Significantly different from the observed value ($p < 0.05$) as determined with the *t* test, indicating antagonistic interactions of herbicides applied in tank mixtures.

### 3.2. Effect of Non-Ammonium Sulfate Water Conditioner on Dicamba and Glyphosate Tank Mixture Interactions (Study 2)

Barnyardgrass presented significant interactions of dicamba by glyphosate ($p = 0.0005$ and $f = 3.6384$) and glyphosate by non-AMS WC ($p < 0.0001$ and $f = 22.7831$). Two two-way ANOVA tables were elaborated, and means were averaged over the respective interaction treatments. For GR horseweed, non-AMS WC presented no significant effect on the biomass reduction of GR horseweed, and the interaction between glyphosate and dicamba was significant ($p < 0.0001$ and $f = 4.6048$). Therefore, means were pooled over dicamba by glyphosate treatments and data was presented as a two-way ANOVA table.

#### 3.2.1. Barnyardgrass

Similar to what was observed in the first study, dicamba was not effective on barnyardgrass, with biomass reduction ranging from 12% to 15% (Table 4). Further, glyphosate rates above 630 g ae ha$^{-1}$ resulted in >83% barnyardgrass biomass reduction. In addition, at 316 g ae ha$^{-1}$ of glyphosate, barnyardgrass biomass reduction was 66%, and the addition of dicamba at any rate decreased biomass reduction. Gowing's correlation demonstrated 16% to 20% decrease in biomass reduction when observed values were compared to expected values.

The addition of non-AMS WC to the spray solution increased glyphosate efficacy on barnyardgrass at 316 and 630 g ae ha$^{-1}$ rates by 26% and 8% when compared to the same glyphosate rates without water conditioner, respectively (Table 4). Such an effect was most likely observed due to the hardness of the water from that region. However, at 1260 g ae ha$^{-1}$ of glyphosate, there was no difference between the addition and not of non-AMS WC, likely due to the increased sensitivity of barnyardgrass to glyphosate; this higher rate without water conditioner was enough to both neutralize the cations present in the solution and control barnyardgrass.

#### 3.2.2. Glyphosate-Resistant Horseweed

The response of GR horseweed to dicamba by glyphosate treatments was similar to what was observed in the first study; when dicamba or glyphosate was applied alone, biomass reduction ranged from 57% to 76%, and from 12% to 31%, respectively (Table 5). The addition of non-AMS WC did not influence GR horseweed biomass reduction by dicamba and glyphosate alone or in tank mixture. In the absence of non-AMS WC, GR

horseweed biomass reduction was 62% compared to 60% when water conditioner was added to treatment solutions (data not shown).

**Table 4.** Effect of water conditioner on dicamba and glyphosate tank mixtures on barnyardgrass biomass reduction (observed and expected) at 28 days after treatment (DAT). Herbicides were applied on 15 cm tall plants. Effects of dicamba by glyphosate and glyphosate by non-AMS WC interactions were significant at $\alpha$ = 0.05; therefore, two two-way tables were elaborated, and means were averaged over the respective interaction treatments.

| Dicamba Dose (g ae ha$^{-1}$) | Glyphosate Dose (g ae ha$^{-1}$) [a,b,c] | | | | | | | |
|---|---|---|---|---|---|---|---|---|
| | 0 | | 316 | | 630 | | 1260 | |
| | Obs | Exp | Obs | Exp | Obs | Exp | Obs | Exp |
| | % of Biomass Reduction | | | | | | | |
| 0 | - | - | 66 Ab | - | 85 a | - | 89 a | - |
| 140 | 15 d | - | 55 Abc | **71 *** | 86 b | 87 | 95 a | 91 |
| 280 | 12 c | - | 52 Bb | **70 *** | 86 a | 87 | 94 a | 91 |
| 560 | 14 c | - | 46 Bb | **66 *** | 83 a | 87 | 94 a | 92 |
| Water conditioner | | | | | | | | |
| No WC | 13 d | | 41 Bc | | 82 Bb | | 93 a | |
| With WC | 14 c | | 67 Ab | | 88 Aa | | 93 a | |

[a] Different letters indicate significant differences ($p$ < 0.05) according to Tukey's least-significant-difference test. Uppercase letters represent differences between treatments in the same column (vertical) and lowercase letters indicate differences between treatments in the same row (horizontal). [b] Expected value obtained from the Gowing's equation: $E = (X + Y) - (XY/100)$, where E is the expected percentage control with dicamba and glyphosate in tank mixture, X and Y are the observed percent control with dicamba or glyphosate alone, respectively. [c] Abbreviations: Obs, observed biomass reduction; Exp, expected biomass reduction according to Gowing's equation; WC, water conditioner; AMS, ammonium sulfate. * Significantly different from the observed value ($p$ < 0.05) as determined with the $t$ test, indicating antagonistic interactions of herbicides applied in tank mixtures.

**Table 5.** Effect of non-AMS water conditioner on dicamba and glyphosate tank mixtures on glyphosate-resistant horseweed biomass reduction (observed and expected) at 28 days after treatment (DAT). Herbicides were applied on 10 cm diameter plants. Effect of water conditioner was not significant at alpha = 0.05. Therefore, means were averaged over dicamba by glyphosate treatments.

| Dicamba Dose (g ae ha$^{-1}$) | Glyphosate Dose (g ae ha$^{-1}$) [a,b,c] | | | | | | | |
|---|---|---|---|---|---|---|---|---|
| | 0 | | 316 | | 630 | | 1260 | |
| | Obs | Exp | Obs | Exp | Obs | Exp | Obs | Exp |
| | % of Biomass Reduction | | | | | | | |
| 0 | - | - | 12 Cb | - | 27 Ca | - | 31 Da | - |
| 140 | 57 Cb | - | 66 Ba | 62 | 64 Bab | **69 *** | 62 Cab | **70 *** |
| 280 | 68 Bc | - | 74 Aa | 72 | 74 Aab | 77 | 69 Bbc | **78 *** |
| 560 | 76 A | - | 77 A | 79 | 77 A | **83 *** | 79 A | **84 *** |

[a] Different letters indicate significant differences ($p$ < 0.05) according to Tukey's least-significant-difference test. Uppercase letters represent differences between dicamba rates at the same glyphosate dose (vertical) and lowercase letters indicate differences between glyphosate rates at the same dicamba dose (horizontal). [b] Expected value obtained from the Gowing's equation: $E = (X + Y) - (XY/100)$, where E is the expected percent control with dicamba and glyphosate in tank mixture, X and Y are the observed percentage control with dicamba or glyphosate alone, respectively. [c] Abbreviations: Obs, observed biomass reduction; Exp, expected biomass reduction according to Gowing's equation; AMS, ammonium sulfate. * Significantly different from the observed value ($p$ < 0.05) as determined with the $t$ test, indicating antagonistic interactions of herbicides applied in tank mixtures.

Antagonistic nature between dicamba and glyphosate was consistently observed when glyphosate was applied at 630 and 1260 g ae ha$^{-1}$, and biomass reduction decreased up to 9% when compared to respective expected values.

## 4. Discussion

The level of GS and GR horseweed control by dicamba or glyphosate alone was similar to what was reported by previous studies. Kumar et al. [29], while investigating a GR horseweed population from Montana, reported that glyphosate at 1260 g ae ha$^{-1}$ resulted in 80% and 40% control of GS and GR horseweed, respectively. Further, in the same study, the authors observed 82% to 98% control from dicamba applications at 560 g ae ha$^{-1}$ for all tested populations. Sikkema et al. [30] reported up to 95% of barnyardgrass control with glyphosate at 900 g ae ha$^{-1}$. Unsatisfactory barnyardgrass control by dicamba alone was expected in the present study due to the reduced activity of this herbicide on grass species.

The current guidelines for best management practices for herbicide-resistant weed control are still heavily reliant on herbicide applications and the use of different modes of action to mitigate the evolution of certain weed biotypes [31,32]. Recent studies recommended DpG tank mixture applications for broad-spectrum weed control and glyphosate-resistant weed management [33–35]. However, it is necessary to understand the nature of the interactions between these herbicides to maximize their efficacy. Antagonistic relations between dicamba and glyphosate have been largely documented in grass types, such as johnsongrass [12], Italian ryegrass (*Lolium perenne*) [36], broadleaf signalgrass (*Urochloa platyphylla*) [36], and giant foxtail (*Setaria faberi*) [37]. Despite the high susceptibility of the barnyardgrass population tested in the present study to glyphosate, reduced control in response to the addition of dicamba is in accordance with the literature. In a similar study, Perkins et al. [38] reported that glyphosate alone at 870 g ae ha$^{-1}$ resulted in 75% of jungle-rice (*Echinochloa colona* (L.) Link) control, and the addition of dicamba at 560 g ae ha$^{-1}$ reduced control to 56%. Moreover, the aforementioned study reported that increments of dicamba at a fixed glyphosate dose consistently decreased jungle-rice control. Such interactions could be especially problematic in agronomic fields with higher grass pressure, where the addition of dicamba to the tank mixture would likely decrease glyphosate efficacy.

Gowing's correlation procedures confirmed antagonistic relations in DpG tank mixtures for GS and GR horseweed control. The findings of our study agree with previous research that reported antagonistic interactions in several broadleaf species, such as kochia (*Bassia scoparia* (L.) A. J. Scott) [11], velvetleaf (*Abutilon theophrasti Medik.*) [9], hemp sesbania (*Sesbania herbacea (Mill.) McVaugh*), prickly sida (*Sida spinosa* L.), pitted morningglory (*Ipomoea lacunosa* L.) [39], and giant ragweed (*Ambrosia trifida* L.) [10]. However, previous research has also reported an increase in weed control when dicamba and glyphosate were applied in tank mixture compared to these herbicides alone, which suggests that this relation might be species-specific. For example, Spaunhorst and Bradley [6] observed that dicamba at 560 g ai ha$^{-1}$ resulted in a 46% biomass reduction of glyphosate-resistant waterhemp (*Amaranthus tuberculatus (Moq.) J. D. Sauer*), compared to 71% when glyphosate at 860 g ae ha$^{-1}$ was added to the tank mixture. Furthermore, Flint and Barrett [40] reported that dicamba and glyphosate absorption in field bindweed (*Convolvulus arvensis* L.) was higher when these herbicides had been applied in tank mixture than that when applied alone.

A more detailed study conducted by Ou et al. [11] utilizing $^{14}$C dicamba and $^{14}$C glyphosate reinforced that, when dicamba and glyphosate were applied in a tank mixture, the absorption of both herbicides was increased when compared to individual application. Nonetheless, glyphosate translocation was reduced throughout the entire length of the study while dicamba translocation was reduced only at later time points. Furthermore, the authors concluded that, although a higher rate of dicamba and/or glyphosate may mask the antagonism, the translocation of both herbicides was reduced when applied to a tank mixture, which compromised herbicidal efficacy. In a similar study, Huff [39] utilized $^{14}$C-dicamba to study the interactions of DpG tank mixtures in the common sicklepod (*Senna obtusifolia* (L.) *H.S. Irwin and Barneby*); consistent with previously reported data, dicamba absorption was higher in the presence of glyphosate, but the translocation of $^{14}$C-dicamba was reduced. Additionally, Merritt et al. [36], while studying the effects of auxin herbicides with glyphosate or clethodim applied in tank mixture or in split applications

on barnyardgrass, observed reduced concentrations of glyphosate in the shoots when dicamba or 2,4-D was applied in tank mixture. However, by applying glyphosate 30 s before 2,4-D or dicamba, glyphosate concentrations in the shoots were similar to those of glyphosate applied alone. Dicamba is a synthetic auxin that can cause metabolic and physical responses in the plant within hours of its application; such effects are responsible for limiting or inhibiting plant growth [41]. Conversely, glyphosate is a slower-acting herbicide that is transported by the phloem to its site of action [42]. Sap movement through the phloem can be affected by dicamba, which may result in a reduced glyphosate translocation and consequently lower efficacy. Further, glyphosate acts within the plant by inhibiting the EPSP enzyme, which shuts down the shikimate pathway [43] and leads to stunt plant growth and restrict phloem movement within the plant. Thus, as dicamba is also transported through the phloem [44], its translocation can be reduced due to the plant's physiological response to glyphosate. The putative mechanism responsible for the antagonism observed between dicamba and glyphosate can be classified as biochemical antagonism in which a chemical decreases the amount of herbicide that reaches the site of action by either reducing absorption, penetration, or transport either by enhancing metabolic inactivation or sequestering, as defined by Green (1989) [45].

In a study conducted by Polli et al. [46] using hard water, the addition of non-AMS WC to DpG tank mixture did not change horseweed and barnyardgrass control, but increased velvetleaf and common lambsquarters control, compared to treatment solutions without water conditioner. The aforementioned study suggests the relation between DpG and non-AMS WC on weed control is also species-specific. Additionally, Zollinger et al. [23] demonstrated that, in 1000 ppm hard water, glyphosate and non-AMS WC presented a higher control of 12 species, including barnyardgrass, than glyphosate alone did. As non-AMS WCs are relatively new in agriculture, there is a lack of available data in the literature about its interactions with dicamba, glyphosate, and other commonly used herbicides.

## 5. Conclusions

The principles of using multiple MOAs in a single application are to mitigate the evolution of herbicide-resistant weeds and improve herbicide efficacy. In general, results observed in this study suggest that efficacy of DpG tank mixtures on barnyardgrass and GS and GR horseweed might be impaired by antagonistic interactions between those herbicides. Split applications could potentially minimize the observed DpG antagonism in specific scenarios. The addition of non-AMS water conditioner improved glyphosate efficacy on barnyardgrass, but the antagonistic DpG relations were sustained in the second part of this study. Ultimately, understanding the nature of the interaction between two or among more herbicides is important to improve weed management and to best use inputs.

**Author Contributions:** E.G.P.: study conception and design, project administration, and draft manuscript preparation. L.H.S.G.: conduction of investigation and data collection. J.H.S.d.S.: analyses and interpretation of results, data curation, and draft manuscript preparation. G.K.: funding acquisition, supervision, and data validation. All authors have read and agreed to the published version of the manuscript.

**Funding:** This research received no external funding.

**Data Availability Statement:** The data presented in this study are available on request from the corresponding author.

**Acknowledgments:** The authors thank the PAT Lab staff and students for their assistance in performing these experiments.

**Conflicts of Interest:** The authors declare no conflict of interest.

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
