# Peer review of "Antagonistic Interactions between Dicamba and Glyphosate on Barnyardgrass (Echinochloa crus-galli) and Horseweed (Erigeron canadensis) Control"

_agronomy, doi:10.3390/agronomy12122942_

Round 1

Reviewer 1 Report

Dear Authors,

I wanted to commend you on your manuscript titled "Antagonistic Interactions between Dicamba and Glyphosate on Barnyardgrass (Echinochloa crus-galli) and Horseweed (Erigeron canadensis) Control." I believe the design and carry-out of the trial was done effectively, and the manuscript reflects that. Since every manuscript can be improved (none are ever perfect) - I have a recommendation to improve the overall impact of the manuscript as presented. I recommend digging a bit deeper into the Ou et al. 2018 paper (citation #11) and the Merritt et al. 2021 (citation # 36) paper in your discussion. WHY was there antagonism observed in the barnyardgrass, and what can you find in those two papers as to better explain the physiological response to the two herbicides. You briefly touch on it in the discussion, but I think based on your results, you can link what you observed and what those authors found to make a more impactful discussion and conclusion point in the paper. This may also help to explain why the non-AMS WC had little effect at the higher glyphosate rates, as this was a central part of the research that went into the planning and design of the experiment.

Overall this is a good manuscript. But I think with a little more explanation in your discussion, it can become a great manuscript - and be one to proudly have on your CV for years to come.

Best Regards,

A. Reviewer

Author Response

To the Agronomy MDPI Reviewer:

Thank you for giving us the opportunity to submit a revised draft of our manuscript. We appreciate the time and effort that you have dedicated to providing your valuable feedback on our manuscript. We were able to improve the discussion of our manuscript to the best of our ability. We appreciate your constructive criticism.  

Best regards,

Estefania G. Polli  

Reviewer 2 Report

Dear Author,

I have gone through your manuscript. It's really very interesting and time demanding. I have the following few observations that should be corrected before it's final acceptance.

P1 L22-24: Not clear. Please revise the sentences. 1 and 2 should not be in brackets.
P2 L85-86: Not clear. Please revise the sentences.

P3 L142: Please replace 'ml' by 'mL'

P8:  In Table 5, the ................%.............. in the 3rd row not clear. Please revise the table.

P9 L 261, 265, 280, 291, 294, 297, 301, 306 & P10 L322, 323, 327: Please follow the correct way to cite the references in the mainbody of the manuscript as per MDPI.

P9 L289-291: Better to write the Scientific name along with the common name of 'kochia' 'velvetleaf' and 'giant ragweed' as like other weeds.

P10 L 339: Better to avoid any citation in the Conclusion section. Conclusion should absolutely be based on your own findings. You have to summarise your findings here. Here, you cannot create any arguments or justify your findings.  

P10-12 L339-471: Please use the MDPI agronomy style in lsiting references.

Thank you

The reviewer

Author Response

To the Agronomy MDPI Reviewer:

Thank you for giving us the opportunity to submit a revised draft of our manuscript. We appreciate the time and effort that you have dedicated to providing your valuable feedback on our manuscript. We were able to incorporate changes to reflect most of the suggestions provided by you.

Best regards,

Estefania G. Polli